# Safety of inadvertent administration of live zoster vaccine to immunosuppressed individuals in a UK-based observational cohort analysis

Daniel J Grint,[1] Helen I McDonald [ORCID],[1] Jemma L Walker,[1,2] Gayatri Amirthalingam,[3] Nick Andrews,[1,2] Sara Thomas[1]

[1]Department of Infectious Disease Epidemiology, London School of Hygiene and Tropical Medicine, London, UK
[2]Statistics, Modelling and Economics Department, Public Health England, London, UK
[3]Immunisation and Countermeasures Division, National Infection Service, Public Health England, London, UK

**Correspondence to**
Dr Helen I McDonald;
helen.mcdonald@lshtm.ac.uk

## ABSTRACT

**Objectives** To investigate the safety of live attenuated varicella zoster vaccination when administered to immunosuppressed individuals.

**Design** Prospective observational cohort study.

**Setting** The study used anonymised data from the Clinical Practice Research Datalink (CPRD), comprising a representative sample of routinely collected primary care data in England between 2013 and 2017 and and linked Hospital Episode Statistics data.

**Participants** 168 767 individuals age-eligible for varicella zoster vaccination registered at a general practice in England contributing data to CPRD.

**Main outcome measures** Electronic health records indicating immunosuppression, zoster vaccination, diagnoses of specific varicella-zoster virus (VZV)-related disease and non-specific rash/encephalitis compatible with VZV-related disease.

**Results** Between 1 September 2013 and 31 August 2017, a period of immunosuppression was identified for 9093/168 767 (5.4%; 95% CI: 5.3%–5.5%) individuals age-eligible for zoster vaccination. The overall rate of vaccination while immunosuppressed was 1742/5251 (33.2 per 100 adjusted person years at risk; 95% CI: 31.9%–34.5%). Follow-up of the 1742 individuals who were inadvertently vaccinated while immunosuppressed identified only two cases of VZV-related disease within 8 weeks of vaccination (0.1%; 95% CI: 0.01%–0.4%), both primary care diagnoses of 'shingles', neither with a related hospital admission.

**Conclusions** Despite evidence of inadvertent vaccination of immunosuppressed individuals with live zoster vaccination, there is a lack of evidence of severe consequences including hospitalisation. This should reassure primary care staff and encourage vaccination of mildly immunosuppressed individuals who do not meet current thresholds for contraindication. These findings support a review of the extent to which live zoster vaccination is contraindicated among the immunosuppressed.

## INTRODUCTION

Herpes zoster (shingles) is a common and painful disease caused by reactivation of varicella-zoster virus (VZV), with debilitating complications including post-herpetic neuralgia. Live-attenuated zoster vaccine was

## Strengths and limitations of this study

► This study investigated the safety of live zoster vaccination during immunosuppression in a large national cohort using electronic health records.
► It is the first study to cover the full profile of causes of immunosuppression listed as contraindications to vaccination in UK national guidance, ascertained from multiple primary and secondary care sources.
► Both primary and secondary care records were used for thorough ascertainment of varicella-zoster virus-related disease, including a sensitivity analysis ascertaining non-specific rash or encephalitis of unspecified aetiology.
► Vaccination rates were analysed using only year of birth for age-eligibility, and so the denominator was adjusted for birth cohorts with partial eligibility.
► Immunosuppression was not distinguished according to severity, but clinicians may have vaccinated selectively and caution would be needed in applying these findings outside of current vaccination practice.

introduced for immunocompetent adults aged 70–79 years in England in 2013, delivered in primary care.[1] The herpes zoster vaccination programme in England was found to have a population impact equivalent to approximately 17 000 fewer episodes of herpes zoster and 3300 fewer episodes of postherpetic neuralgia among 5.5 million eligible individuals in the first 3 years of the programme.[2]

Immunosuppression is associated with a high burden of zoster and its complications,[3 4] and there have been calls to consider vaccination for this population.[5] However, live zoster vaccine is currently contraindicated in immunosuppression as it may cause VZV-related disease.[1] High-profile case reports of fatal vaccine-related disease among severely immunosuppressed individuals have caused concern and may have contributed to declining vaccine coverage.[6 7] Understanding

the safety of live vaccination during immunosuppression is important to support guidance on use of the vaccine, to ensure that individuals who can safely benefit from the vaccine are enabled to do so.

A new vaccine which is recombinant rather than containing live virus could offer a safer alternative for immunosuppressed individuals without the risk of vaccine-related disease, and has been found to be effective among patients with autologous haematopoietic stem cell transplantation.[8] However, supplies are currently unable to meet global demands. The Joint Committee on Vaccination and Immunisation has recommended use of the recombinant vaccine for individuals with immunosuppression. Understanding the safety of live vaccination for typical causes of immunosuppression will be important to prioritise use of limited supplies of the new recombinant vaccine. When recombinant vaccine is used in cases for which live vaccine is contraindicated, this may introduce potential for confusion and it will be even more important to understand the consequences of inadvertent live vaccination during immunosuppression.

This study aimed to investigate the frequency and consequences of live zoster vaccination during immunosuppression among a large UK cohort from 2013 to 2017.

## METHODS

### Data source

This study used anonymised data from the Clinical Practice Research Datalink (CPRD). The data include information on year of birth, medical diagnoses (version 2 Read codes), prescriptions and vaccinations. For 60% of individuals, records are prelinked to anonymised hospitalisation data (Hospital Episode Statistics, HES). HES-linked data for inpatient admissions (International Classification of Diseases, ICD-10 codes) and procedures (OPCS-4 Classification of Interventions and Procedures codes) were used to supplement identification of immunosuppressed individuals and VZV-related disease.

### Study population

Immunosuppressed individuals age-eligible for zoster vaccination, active in CPRD from September 2013 to August 2017 and registered with a CPRD practice for at least a year before study entry, were included.

Age eligibility for zoster vaccination has differed each year since the vaccination introduction. As month of birth was not available, individuals born in years for which ≥67% of the population would have been eligible for vaccination were included (online supplementary appendix A1). Birth cohorts were defined as 'maiden cohorts' in the first year for which they were age-eligible for vaccination.

Periods of immunosuppression were identified using Read codes and prescription records from CPRD, plus ICD-10 codes and OPCS codes in linked HES data using algorithms previously described.[9] Immunosuppression was defined based on contraindications to live zoster vaccination described in national guidance.[1] The time

periods assigned to each immunosuppressing condition or medication type, and dose thresholds for relevant medications, are described in online supplementary appendix A2. For prescription records missing dose, the median was imputed according to category of age and sex, in line with previous zoster studies.[4]

### Vaccination status and VZV-related disease

Individuals were followed to the first positive record of zoster vaccination. If this indicated that the vaccine was delivered by another healthcare provider the individual was excluded from the cohort, as timing of vaccination could not be determined (n=29).

Evidence of VZV-related disease in primary or secondary care records was assessed during the 8 weeks following a vaccination given while immunosuppressed. For the primary analysis, only specific diagnoses of VZV disease were included. Sensitivity analysis also included any rash that was unspecified or compatible with VZV, and acute encephalitis of unspecified aetiology. For individuals with HES-linkage, any diagnosis recorded within 8 weeks following vaccination was used to supplement identification of VZV-related disease. Codelists are available at https://datacompass.lshtm.ac.uk/1336/

### Statistical methods

An open prospective cohort study design was used whereby individuals exited and re-entered the cohort according to time-varying immunosuppression status. Follow-up started on 1st September of the study year in which the individual was age-eligible for vaccination and ended at the earliest of death, date of transfer out of practice, practice last collection date, elapsed age-eligibility for zoster vaccination, resolution of immunosuppression or 31 August 2017.

The vaccination rate was calculated by total person years at risk (PYAR) while immunosuppressed with adjustment to account for age-eligibility uncertainty from unknown month of birth (online supplementary appendix A1). Cumulative uptake of zoster vaccine was computed stratified by treatment cohort and programme year.

The number of vaccinated immunosuppressed individuals who developed VZV-related disease in the subsequent 8 weeks was described. In sensitivity analysis, disease in the first week following vaccination was excluded.

Statistical analysis was performed using STATA V.14.2 and SAS V.9.4.

### Patient and public involvement

Patients and the public were not involved in the design or planning of the study.

## RESULTS

Between 1 September 2013 and 31 August 2017 data were available for 168 767 individuals age-eligible for vaccination, of whom 89 416 (53.0%) were female and 76 337 (45.2%) in the catch-up cohort. A period of immunosuppression while age-eligible for vaccination was identified for 9093/168 767 (5.4%; 95% CI: 5.3%–5.5%).

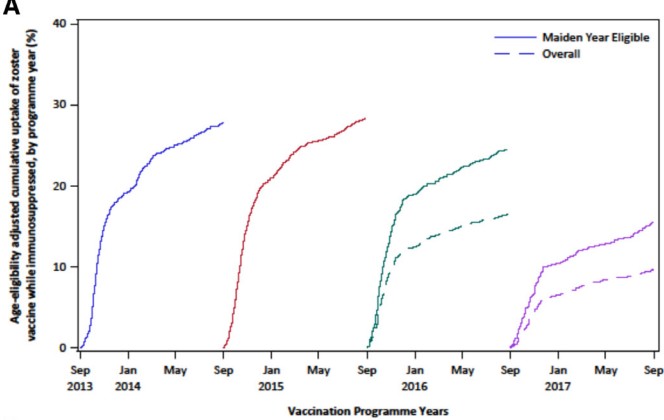

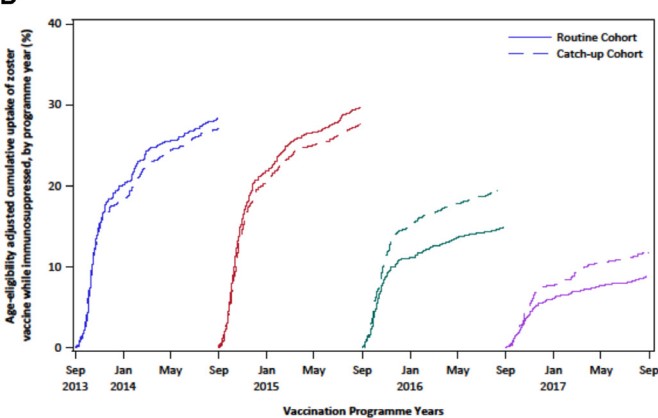

**Figure 1** Cumulative uptake of zoster vaccination while immunosuppressed, by vaccination programme year. (A) Stratified by maiden years of eligibility or overall eligibility. (B) Stratified by routine or catch-up cohort.

One thousand seven hundred forty-two individuals were vaccinated during a period of immunosuppression. Adjusting PYAR while immunosuppressed to account for age-eligibility uncertainty, the overall rate of vaccination during immunosuppression was 1742/5251 (33.2 per 100 adjusted PYAR; 95% CI: 31.9–34.5). Figure 1 shows the cumulative uptake of zoster vaccine by programme year overall, in maiden years of eligibility and by cohort. Cumulative uptake was higher in programme years 3 and 4 when restricted to maiden years of eligibility. Cumulative uptake was highest in programme years 1 and 2 for both the routine and catch-up cohorts.

Among those vaccinated while immunosuppressed, the most common underlying cause was chemotherapy (55.3%), followed by other immunosuppressant therapies including biologics (13.2%), multiple indications (11.4%) and steroid drug use (11.3%). Forty seven (2.7%) had a permanent cause of immunosuppression; 368 (21.1%) were immunosuppressed for the duration of follow-up; median follow-up 32.2 months (IQR: 19.7–48.0). Vaccination took place during the final 4 weeks of a defined period of immunosuppression for 138/1742 (7.9%).

In the 8 weeks following vaccination, 2/1742 (0.1%; 95% CI: 0.01–0.4%) had a diagnosis of 'shingles' recorded in primary care. Both individuals had HES-linkage

available; however, neither had a related hospital admission. One of these cases occurred within 7 days of vaccination.

Using a broader definition including non-specific rash or encephalitis identified a further 23 possible cases of VZV-related disease (in total 25/1742 (1.4%; 95% CI: 0.9%–2.1%)). All of these were instances of unspecified rash in primary care, and there were no cases of encephalitis. In total, 22/25 (88%) had HES-linkage available; however, none had a related hospital admission recorded. Five of the broader definition cases occurred within 7 days of vaccine administration.

Chemotherapy was the cause of immunosuppression for the majority of cases who developed specific or non-specific potentially vaccine-related disease (15/25 (60%)). The remaining cases included immunosuppression by oral steroid use, other immunosuppressant medications, leukaemia and organ transplant.

## DISCUSSION

This is the first study to investigate the safety of live zoster vaccination across the range of contraindicating immunosuppressive conditions. Our study identified 1742 individuals vaccinated while immunosuppressed and two subsequent cases with a diagnosis of 'shingles', with no related hospitalisations, and no cases of encephalitis.

A key strength of this study is the thorough ascertainment of both immunosuppression and VZV-related disease using linked primary and secondary care data for a large, representative cohort with a range of immunosuppressive conditions.

The study has limitations. A key limitation is that month of birth was not available for precise identification of age-eligibility. If immunosuppressed individuals in a birth cohort with 67% eligibility were vaccinated while not age-eligible, rates of vaccination in immunosuppression would be overestimated. There also remains uncertainty in defining time-periods of immunosuppression and in imputing missing dose data for medications which may result in underascertainment or overascertainment of immunosuppression.

It is possible that VZV-related disease may have been underascertained, either because patients did not attend healthcare or due to non-specific coding. A US study reported that 95% of patients aged over 60 years had attended healthcare when they experienced zoster disease,[10] and this might be expected to be higher among immunosuppressed individuals in a setting with universal healthcare. This study used both primary care and secondary care records to ascertain VZV-related disease, an approach which has previously been found to generate plausible estimates of zoster incidence among the older general population.[11] The sensitivity analysis was also designed to ascertain possible cases of VZV-related disease which may have been coded non-specifically as rash or encephalitis. Conversely, as this population has a high baseline risk of naturally occurring shingles, we may have

over-estimated the risk attributable to vaccination, particularly when including cases within 7 days of vaccination.

Finally, while our definitions of immunosuppression followed national guidance, we could not replace clinical judgement on severity or timing of immunosuppression, and the study was not powered to assess safety according to type of immunosuppression.[1] Clinicians may have selectively, rather than inadvertently, vaccinated individuals at lower risk of vaccine-related disease, resulting in a 'healthy vaccinee' effect, and caution would be needed in generalising these findings outside of current practice in the context of guidance on contraindications. However, the most frequent contraindication was chemotherapy, and vaccinations did not occur disproportionately towards the end of a period of immunosuppression, suggesting that vaccinations were not all at the margins of the guidance.

Our safety findings are consistent with previous studies of live zoster vaccination among patients with selected immunosuppressive conditions.[12–14] Studies showing that VZV-specific immunity may persist or even be boosted by vaccination during cell-mediated immunosuppression,[15–17] further support the plausibility of residual immunity against vaccine-related disease despite immunosuppression.

Rates of zoster vaccination during immunosuppression were high, and similar in routine and catch-up cohorts. To our knowledge, this is the first study to calculate rates of live zoster vaccination across this range of immunosuppressing conditions. A zoster vaccine effectiveness study of Medicare beneficiaries in the USA included 140 925 individuals with a diagnosis of leukaemia, lymphoma or HIV, of whom 4469 (3.2%) were vaccinated while immunosuppressed, comparable to the overall study vaccine uptake (29 785/766 330; 3.9%), suggesting that live zoster vaccination despite immunosuppression is not unique to the UK setting.[16] In our study, analysis restricted to maiden years of eligibility suggests that the apparent decline in vaccination rates after year 2 is partly a cohort effect, whereby people who were unvaccinated despite previous eligibility were less likely to be vaccinated subsequently. This could be due to an initial decision not to vaccinate continuing over subsequent years of eligibility, or a greater focus on vaccination for newly eligible patients. Increasingly detailed guidance over time may also have helped reduce inadvertent vaccination.

Among this large cohort with a range of immunosuppressing conditions, we found no evidence of severe VZV-related disease following live zoster vaccination while immunosuppressed. This should reassure clinicians, and encourage vaccination of mildly immunosuppressed individuals who do not meet current thresholds for contraindication, especially in the current context of declining uptake of the national programme.[18] These findings support a review of the extent to which live zoster vaccination is contraindicated among the immunosuppressed population. Further research is needed to identify any patient groups for whom recombinant zoster vaccine should be prioritised once stocks become available in the UK.

**Contributors** SLT had the idea for the study. SLT, JLW, GA and NA obtained the data. All authors contributed to the design of the study. DJG analysed the data. All authors contributed to the interpretation of results. DJG and HIM drafted the manuscript and all authors revised it critically. All authors approved the final version to be published and agree to be accountable for all aspects of the work. DJG is the guarantor. The corresponding author attests that all listed authors meet authorship criteria and that no others meeting the criteria have been omitted.

**Funding** The research was funded by the National Institute for Health Research (NIHR) Health Protection Research Unit (HPRU) in Immunisation at the London School of Hygiene and Tropical Medicine in partnership with Public Health England (PHE).

**Competing interests** DJG, JLW, NA and SLT had financial support from the National Institute for Health Research (NIHR) Health Protection Research Unit (HPRU) in Immunisation for the submitted work; the Public Health England Immunisation Department has provided vaccine manufacturers with post-marketing surveillance reports on pneumococcal and meningococcal infection which the companies are required to submit to the UK Licensing Authority in compliance with their Risk Management Strategy, and a cost recovery charge is made for these reports.

**Patient consent for publication** Not required.

**Ethics approval** The study was approved by the Independent Scientific Advisory Group of the CPRD (ISAC reference 18_218A) and the London School of Hygiene and Tropical Medicine Ethics Committee (reference 16298). The amended ISAC protocol was made available to reviewers.

**Provenance and peer review** Not commissioned; externally peer reviewed.

**Data availability statement** Data may be obtained from a third party and are not publicly available. These data were obtained from the Clinical Practice Research Datalink, provided by the UK Medicines and Healthcare products Regulatory Agency. The authors' licence for using these data does not allow sharing of raw data with third parties. Information about access to Clinical Practice Research Datalink data is available here: https://www.cprd.com/research-applications. Codelists for this study are available at https://datacompass.lshtm.ac.uk/1336/

**ORCID iD**
Helen I McDonald http://orcid.org/0000-0003-0576-2015

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
