## [Reviewer comments · BMJ Open]

ARTICLE DETAILS

TITLE (PROVISIONAL)	Safety of inadvertent administration of live zoster vaccine to immunosuppressed individuals in a UK-based observational cohort analysis
AUTHORS	Grint, Daniel; McDonald, Helen; Walker, Jemma; Amirthalingam, Gayatri; Andrews, Nick; Thomas, Sara

VERSION 1 – REVIEW

REVIEWER	Steven Pergam, MD, MPH Fred Hutchinson Cancer Research Center Seattle, WA USA I have participated in clinical trials with Merck & Co, Chimerix, Inc and receive research support from Global Life Technologies.
REVIEW RETURNED	20-Oct-2019

GENERAL COMMENTS	Dr. Grint and colleagues present this manuscript which evaluates inadvertent administration of live-virus Oka strain zoster vaccine in immunosuppressed patients. Using a large administrative dataset available in the UK, they evaluated 168,767 patients and identified over 9,000 age eligible for HZ vaccine who were considered to be IC hosts by reviewing ICD-9, pharmacy codes, and other sources. A total of 1742 individuals were vaccinated during a period of immunosuppression, and through administrative data review the authors only found 2 “proven” HZ events in the 8 weeks following vaccination, suggesting more limited risk amongst those reviewed. 1) It would be helpful for the reader to see a table of those IC hosts who were vaccinated versus those that were not vaccinated. This would help to assess which groups were targeted (more likely to get vaccinated), versus those who were not targeted (e.g. HCT/SOT recipients). 2) Herpes zoster endpoints based on ICD-9/10 coding have been assessed in prior studies are known to have limitations on data capture (multiple publication noting this weakness). Did the authors consider a sensitivity analysis to review a subset of patients with and without HZ complications to assure accuracy? If not this limitation should be noted in the manuscript. 3) One important characteristic that the authors have not addressed in the use of prophylaxis with acyclovir & other antiviral agents with efficacy against VZV which is commonly used in some high-risk populations. Were any of the patients receiving HZ vaccine receiving acyclovir/valacyclovir during the period at risk? 4) As a follow-up to this prior comment – Acyclovir/Valacyclovir/Famciclovir are the primary agents for treatment of HZ. Could the authors assess acute use of these agents as a sensitivity analysis for HZ events that were missed by ICD-10 coding?
---

	5) How can the authors try to address bias of those receiving HZ vaccine as those who would caregivers might consider lower risk vs. those at high risk not being offered vaccine? This parity is important when considering opportunities for expanding vaccine use in these patients. 6) Did the authors consider evaluating HZ risk among patients who received HZ vaccine compared to those who did not over time? 7) What is highly unusual is that based on these data that nearly 20% of what the authors considered immunosuppressed patients (1742/9093) received the live-virus HZ vaccine against what were considered contraindications by national guidance. Can the authors speculate on why the vaccine was given so frequently? Or does this suggest that their criteria for IC are too broad? Minor 1) I would not use the term “non-steroid drug” (Page 10 line 38-39) as this is could confuse the reader with NSAIDs. I would use other immunosuppressive therapy 2) A weakness to these data are that in the UK HZ vaccine is only approved for patients over 70 year of age. Some of the IC approaches, are less likely to be offered to older patients. Can the authors note this limitation and how this might be different in countries where approval is at a younger age (i.e. North America).
--	---

REVIEWER	Prof Kristine Macartney National Centre For Immunisation Research & Surveillance (NCIRS)
REVIEW RETURNED	10-Nov-2019

GENERAL COMMENTS	This is an elegant and important study, intended to examine any adverse events from varicella zoster virus (VZV) vaccine related disease in immunocompromised recipients of the live attenuated zoster vaccine. Given reports of two fatalities in this context, understanding population level rates of inadvertent administration and associated risks are important. Minor suggestions or comments Abstract  - Overall, note that 'shingles', while an appropriate code to search for in medical records, is not the correct term for vaccine VZV related disease - shingles is reactivation of latent virus from primary infection - the clinical manifestation/s of OkaVZV disease in immunocompromised is not a dermatomally distributed rash. Suggest minor adjustments to wording throughout - Conclusions could also note the high rate of inadvertent administration/non adherence to practice guidelines - to my knowledge this is new finding and could also be compared (in the discussion) to US studies that have examined same if possible? Introduction  -Lines 33-39 could the authors please be slightly more detailed in their description of the potential future parallel use of LA HZ vaccine and recomb HZ vaccine for the general BMJ readership. Also in Discussion. Readers may not be aware that rHZ vaccine is not 'live' and hence poses no risk of vaccine virus associated disease. Possibly also reference to RCT on efficacy in HSCT patients (Sullivan Jama 2019) Methods  - Did authors conduct any comparison to determine if hospitalisation codes in patients with HES data were consistent with Read codes,
--

	or added to number of patients classified with immunosuppression? Results  - pg 10 lines 40-47. Did authors examine the proportion of patients who were vaccinated shortly after starting immunosuppression or before hand? (noting some guidelines allow for later). - how many of the 25 patients with broader definition were rash v encephalitis - it would seem odd that patients with encephalitis didn't have a related hospital admission - the overall rate of HZ in this population appears low - possibly lower than expected in the context of a vaccine with 50% effectiveness - which authors acknowledge in Discussion. To validate case ascertainment methods, can the authors comment, perhaps in Discussion, on whether the rate of HZ was comparable to that in effectiveness studies, eg either in the first 2 weeks (noting could be 'well vaccinee effect' there, or possibly in a control period later after vaccination? - pg 11, line 13, suggest say non-specific 'potentially' vaccine related disease - also noting in the discussion, that the most frequent contraindication was chemotherapy (? overall or in vaccinated), I wondered if more data in results on rates of inadvertent use by broad category of immunosuppression would be useful? Discussion  pg 13 lines 13-18 and Appendix 1A - its hard to tell which are 'maiden cohort's and which catch up. If 'maiden' have never had vaccine offered to them before, why would they contribute to decline in inadvertent use in later 2 study years. pg 13 line 25 - while I agree, with the statement for clinicians to consider vaccinating 'mildly immunosuppressed' individuals, it is in practice often difficult for primary care GPs/nurses to be sure of what is 'mild'. Are UK PHE changing their guidelines based on this study, or feel they can better define for clinicians what is mild versus not mild (noting current UK guidelines comprehensive but challenge is at individual patient level assessment) pg 13 line 29 - pls provide reference for declining coverage data Appendix A2. Is it possible to list which (presumably many) biologics were included - this is common clinical question and some biologics have only narrow or v. little immunosuppressive effects
--	---

VERSION 1 – AUTHOR RESPONSE

Reviewer 1: Steven Pergam, MD, MPH

Dr. Grint and colleagues present this manuscript which evaluates inadvertent administration of live-virus Oka strain zoster vaccine in immunosuppressed patients. Using a large administrative dataset available in the UK, they evaluated 168,767 patients and identified over 9,000 age eligible for HZ vaccine who were considered to be IC hosts by reviewing ICD-9, pharmacy codes, and other sources. A total of 1742 individuals were vaccinated during a period of immunosuppression, and through administrative data review the authors only found 2 “proven” HZ events in the 8 weeks following vaccination, suggesting more limited risk amongst those reviewed.

1) It would be helpful for the reader to see a table of those IC hosts who were vaccinated versus those that were not vaccinated. This would help to assess which groups were targeted (more likely to get vaccinated), versus those who were not targeted (e.g. HCT/SOT recipients).

Response: We agree it would be useful to investigate more fully who was more likely to be vaccinated while immunosuppressed, to understand the high incidence of vaccination despite contraindicating immunosuppression (as your comment #7 highlights).

Our study was designed to investigate the safety of live zoster vaccination during any contraindicating immunosuppression. We did not distinguish between types of immunosuppression in identifying time periods of immunosuppression, as we would not have had sufficient power to compare vaccine safety by type of immunosuppression.

Due to the combination of immunosuppression patterns with restrictions on age eligibility for the vaccine, some patients have very short periods of time at risk of vaccination and these vary greatly (for example, patients with intermittent steroid prescriptions have very different time periods of immunosuppression to patients who are considered immunosuppressed for two years following a bone marrow transplant). We would therefore wish to take person-time into account if investigating this question.

We have reported the type of immunosuppression that was present at the time of vaccination, as it could be identified within our safety study, and we hope this helps somewhat to understand which immunosuppressed individuals are being vaccinated.

“Among those vaccinated while immunosuppressed the most common underlying cause was chemotherapy (55.3%), followed by other immunosuppressant therapy (13.2%), multiple indications (11.4%), and steroid drug use (11.3%). 47 (2.7%) had a permanent cause of immunosuppression.”

However, to describe rates of immunosuppression according to type of immunosuppression, we would need to redesign the analysis to identify the study population of patients with immunosuppression according to timing of each type of immunosuppression. We agree this would be interesting as a future study, but are afraid it is outside the scope of our current study. We would suggest that it would be best conducted in a larger dataset, for the cohorts of individuals with different types of immunosuppression to be large enough to distinguish type of immunosuppression with reasonable granularity. In the meantime, we have highlighted this as a limitation of the study in the discussion:

“Finally, while our definitions of immunosuppression followed national guidance, we could not replace clinical judgement on severity or timing of immunosuppression, and the study was not powered to assess safety according to type of immunosuppression.(1) Clinicians may have selectively, rather than inadvertently, vaccinated individuals at lower risk of vaccine-related disease, resulting in a ‘healthy vaccinee’ effect, and caution would be needed in generalising these findings outside of current practice in the context of guidance on contraindications. However, the most frequent contraindication was chemotherapy, and vaccinations did not occur disproportionately towards the end of a period of immunosuppression, suggesting that vaccinations were not all at the margins of the guidance.”

2) Herpes zoster endpoints based on ICD-9/10 coding have been assessed in prior studies are known to have limitations on data capture (multiple publication noting this weakness). Did the authors consider a sensitivity analysis to review a subset of patients with and without HZ complications to assure accuracy? If not this limitation should be noted in the manuscript.

Response: Thank you for raising this important point. We believe one of the strengths of this study is that we used Read v2 codes in primary care as well as ICD-10 codes from hospital admission records to ascertain zoster cases, and we included a sensitivity analysis which used a broad definition of rash or acute encephalitis in order to ascertain possible cases of zoster which may have been coded non-specifically. However, we recognise that we may still have missed cases, either because patients did not present to healthcare or due to poor sensitivity of codes. We have expanded the discussion on this point, to read:

“It is possible that zoster disease may have been under-ascertained, either because patients did not attend healthcare or due to non-specific coding. A US study reported that 95% of patients aged over 60 years had attended healthcare when they experienced zoster disease,(7) and this might be expected to be higher among immunosuppressed individuals in a setting with universal healthcare. This study used both primary care and secondary care records to ascertain zoster cases, an approach which has previously been found to generate

plausible estimates of zoster incidence among the older general population.(8) The sensitivity analysis was also designed to ascertain possible cases of zoster which may have been coded non-specifically as rash or encephalitis. Conversely, as this population has a high baseline risk of naturally occurring shingles, we may have over-estimated the risk attributable to vaccination, particularly when including cases within 7 days of vaccination.”

We hope that this more adequately addresses this important issue.

3) One important characteristic that the authors have not addressed in the use of prophylaxis with acyclovir & other antiviral agents with efficacy against VZV which is commonly used in some high-risk populations. Were any of the patients receiving HZ vaccine receiving acyclovir/valacyclovir during the period at risk?

Response: Thank you for this thoughtful suggestion. We did not identify prophylactic antiviral use in the study as we could not identify it fully and would be more likely to miss it for the more immunosuppressed patients. In this dataset, prescription records are available from primary care but not secondary care, and we would therefore miss antiviral prescriptions in secondary care including outpatients, which would be the more likely setting for the most immunosuppressed patients than primary care.

4) As a follow-up to this prior comment – Acyclovir/Valacyclovir/Famciclovir are the primary agents for treatment of HZ. Could the authors assess acute use of these agents as a sensitivity analysis for HZ events that were missed by ICD-10 coding?

Response: Thank you for another thoughtful suggestion. Unfortunately in this dataset, prescriptions are available from primary care but not secondary care and we would therefore miss antivirals prescribed in A&E or as part of inpatient admissions. We would expect immunosuppressed patients with zoster to be largely treated in secondary care with intravenous antivirals. We therefore did not identify acute antiviral use as unfortunately this could not be ascertained in secondary care settings, which would be the most relevant for this patient population.

5) How can the authors try to address bias of those receiving HZ vaccine as those who would caregivers might consider lower risk vs. those at high risk not being offered vaccine? This parity is important when considering opportunities for expanding vaccine use in these patients.

Response: Thank you for raising this important point. We agree that clinicians are likely to have vaccinated selectively, and we could not replace clinical judgement as to who to vaccinate according to severity of immunosuppression. To highlight this important point, we have expanded the discussion to read:

“Clinicians may have selectively, rather than inadvertently, vaccinated individuals at lower risk of vaccine-related disease, resulting in a ‘healthy vaccinee’ effect, and caution would be needed in generalising these findings outside of current practice in the context of guidance on contraindications.”

And have also taken the opportunity to highlight this in the strengths and limitations summary:

“Immunosuppression was not distinguished according to severity, but clinicians may have vaccinated selectively and caution would be needed in applying these findings outside of current vaccination practice.”

Timing of vaccination relative to the course of immunosuppression may have been relevant to clinical judgement, and so we did look at the proportion of patients vaccinated in the final 4 weeks of their immunosuppressed time period, which was not markedly disproportionate.

6) Did the authors consider evaluating HZ risk among patients who received HZ vaccine compared to those who did not over time?

Response: Thank you for this interesting question. We have previously evaluated this among the general population (Walker et al., Effectiveness of herpes zoster vaccination in an older United Kingdom population, *Vaccine* 2018;36(17): 2371-2377)

For this study, we did consider the possible comparison from two perspectives. Firstly, as a potential control group. This is a population with a high baseline risk of zoster, and the approach we took of describing all zoster within 8 weeks of vaccination may include naturally occurring zoster that is not vaccine-related. If vaccinations were all inadvertently at random among immunosuppressed patients,

then a comparator group might have allowed us to distinguish excess cases of zoster as likely to be vaccine-related. However, we were aware that vaccination despite immunosuppression might be selective (both less likely to happen inadvertently among severely immunosuppressed individuals, and perhaps to occur deliberately among selected individuals based on clinical judgement), and also that timing of vaccination was likely to reflect both the course of immunosuppression and any history of or exposure to zoster, and that we were unlikely to be able to adjust for this using available data. The baseline rate of zoster between patients who received vaccine and those who did not might therefore not be the same, and this could have biased our findings in either direction. We decided on the approach we took as although it might slightly overestimate the incidence of vaccine-related zoster by including all cases within 8 weeks of vaccination, from a safety perspective this is the most cautious approach. Secondly, comparing HZ risk over time among patients who received HZ vaccine to those who did not (preferably by type of immunosuppression) might also be useful in a study designed to understand who is receiving the vaccine despite immunosuppression, and why. This was outside the scope of our safety study, but could be interesting in a future study of who is receiving vaccination in a large database, to inform a review of the current guidelines.

7) What is highly unusual is that based on these data that nearly 20% of what the authors considered immunosuppressed patients (1742/9093) received the live-virus HZ vaccine against what were considered contraindications by national guidance. Can the authors speculate on why the vaccine was given so frequently? Or does this suggest that their criteria for IC are too broad?

Response: We agree that a surprisingly high proportion of immunosuppressed patients were vaccinated. There is some inherent uncertainty in our definitions of immunosuppression (for example, there is scope for the imputation of steroid doses for patients with missing dose values to result in misclassification) but this would not explain the majority of the cases, particularly as chemotherapy was the most commonly observed cause of immunosuppression. We may have defined the duration of periods of immunosuppression too generously, although these definitions followed the national guidelines. For some criteria there may be exceptions to contraindication (for example, biological therapies are considered contraindications for 12 months unless otherwise directed by a specialist), but this would apply to a small number of patients.

Clinicians may be using judgement to vaccinate patients selectively – and our findings would suggest that current practice is not cause for concern. Our definitions of immunosuppression followed the national guidance on contraindications for the live zoster vaccine, and we suggest that the findings support a review of the extent to which live zoster vaccination is contraindicated among the immunosuppressed, as there may be scope to relax these thresholds.

UK guidance was also refined during the study period with updates made to the “Green Book” – for example, dosage for drugs like methotrexate were not initially included in guidance, and were added in a 2015 update. So the initial guidance left more to clinical discretion.

We have expanded the discussion of each of these points in the discussion.

Minor

1) I would not use the term “non-steroid drug” (Page 10 line 38-39) as this could confuse the reader with NSAIDs. I would use other immunosuppressive therapy

Response: Thank you for pointing out the potential for confusion. The 13.2% includes individuals receiving biologics, as well as those receiving medications defined in appendix 2 as ‘other immunosuppressive therapies’, and so we have amended the wording to “other immunosuppressant therapies including biologics”, and hope this is clearer.

2) A weakness to these data are that in the UK HZ vaccine is only approved for patients over 70 year of age. Some of the IC approaches, are less likely to be offered to older patients. Can the authors note this limitation and how this might be different in countries where approval is at a younger age (i.e. North America).

Response: Thank you for raising this point – we have added it to the limitations section in our discussion:

“Our study population was 70 years or over, and these findings may also not generalise to younger vaccine recipients, in whom patterns of immunosuppression are likely to differ.”

Reviewer 2: Prof Kristine Macartney

This is an elegant and important study, intended to examine any adverse events from varicella zoster virus (VZV) vaccine related disease in immunocompromised recipients of the live attenuated zoster vaccine. Given reports of two fatalities in this context, understanding population level rates of inadvertent administration and associated risks are important.

Response: Thank you for your kind comments.

Minor suggestions or comments

Abstract

- Overall, note that 'shingles', while an appropriate code to search for in medical records, is not the correct term for vaccine VZV related disease - shingles is reactivation of latent virus from primary infection - the clinical manifestation/s of OkaVZV disease in immunocompromised is not a dermatomally distributed rash. Suggest minor adjustments to wording throughout

Response: Thank you for this helpful improvement to the manuscript. We have changed the wording throughout to refer to cases of zoster, and use “shingles” only in the context of explaining how the two case diagnoses were recorded in primary care.

- Conclusions could also note the high rate of inadvertent administration/non adherence to practice guidelines - to my knowledge this is new finding and could also be compared (in the discussion) to US studies that have examined same if possible?

Response: Thank you for this helpful comment, we have added a discussion as you suggest:
“To our knowledge, this is the first study to calculate rates of live zoster vaccination across a range of immunosuppressing conditions. A zoster vaccine effectiveness study of Medicare beneficiaries in the US included 140,925 individuals with a diagnosis of leukaemia, lymphoma or HIV, of whom 4,469 (3.2%) were vaccinated while immunosuppressed, comparable to the overall study vaccine uptake (29,785/766,330; 3.9%), suggesting that live zoster vaccination despite immunosuppression is not unique to the UK setting.(13)”

Introduction

-Lines 33-39 could the authors please be slightly more detailed in their description of the potential future parallel use of LA HZ vaccine and recomb HZ vaccine for the general BMJ readership. Also in Discussion. Readers may not be aware that rHZ vaccine is not 'live' and hence poses no risk of vaccine virus associated disease. Possibly also reference to RCT on efficacy in HSCT patients (Sullivan Jama 2019)

Response: Thank you for this suggestion to make the context clearer to the general reader. We have added more detail to the explanation (including the RCT reference) as you helpfully suggest:

“A new vaccine which is recombinant rather than containing live virus could offer a safer alternative for immunosuppressed individuals without the risk of vaccine-related zoster, and has been found to be effective among patients with autologous hematopoietic stem cell transplantation (HSCT).(7) However, supplies are currently unable to meet global demands. The Joint Committee on Vaccination and Immunisation has recommended use of the recombinant vaccine for individuals with immunosuppression. Understanding the safety of live vaccination for typical causes of immunosuppression will be important to prioritise use of limited supplies of the new recombinant vaccine. When recombinant vaccine is used in cases for which live vaccine is contraindicated, this may introduce potential for confusion and it will be even more important to understand the consequences of inadvertent live vaccination during immunosuppression.”

Methods

- Did authors conduct any comparison to determine if hospitalisation codes in patients with HES data were consistent with Read codes, or added to number of patients classified with immunosuppression?

Response: We did not explore the value of HES data in defining immunosuppression in this study. We have used these methods previously to define immunosuppression, and found that HES data add to ascertainment of immunosuppression, particularly for patients with chemotherapy. We have clarified that this was not a new algorithm and provided a reference, and hope this is helpful:

“Periods of immunosuppression were identified using Read codes and prescription records from CPRD, plus ICD-10 codes and OPCS codes in linked HES data using algorithms previously described.(8)”

Results

- pg 10 lines 40-47. Did authors examine the proportion of patients who were vaccinated shortly after starting immunosuppression or before hand? (noting some guidelines allow for later).

Response: Thank you, this is an interesting suggestion, and the UK guidelines encourage the latter. As the focus of this study was on the incidence of zoster following inadvertent live vaccination, we did not investigate the period prior to onset of immunosuppression – however, given the high rate of vaccination during immunosuppression, we agree this would warrant investigation in a future study to understand patterns of vaccination prior to and during immunosuppression. As the study follow up criteria combined age eligibility with immunosuppressing events, this study was not designed to allow us to identify the onset of immunosuppression, and so we did not examine timing relative to the start of the period at risk. We did, though, consider timing towards the end of the period at risk as a disproportionate uptake of vaccination at the end of the period at risk might have suggested that the definitions of time at risk were too long. We present this in the results:

“Vaccination took place during the final four weeks of a defined period of immunosuppression for 138/1,742 (7.9%).”

- how many of the 25 patients with broader definition were rash v encephalitis - it would seem odd that patients with encephalitis didn't have a related hospital admission

Response: Thank you for highlighting that our results were not clear as to the absence of any encephalitis cases. We agree that a record of encephalitis without a related hospital admission would have raised concerns about the accuracy of the records. All of the patients who met the broader definition were cases of “unspecified rash” recorded in primary care. We have changed the wording in the results to clarify that there were no cases of encephalitis. It now reads:

“Using a broader definition including non-specific rash or encephalitis identified a further 23 possible cases of VZV disease (in total 25/1,742 (1.4%; 95% CI: 0.9-2.1%). All of these were instances of unspecified rash in primary care, and there were no cases of encephalitis.” (Page 8)

We hope this clarifies this important point.

- the overall rate of HZ in this population appears low - possibly lower than expected in the context of a vaccine with 50% effectiveness - which authors acknowledge in Discussion. To validate case ascertainment methods, can the authors comment, perhaps in Discussion, on whether the rate of HZ was comparable to that in effectiveness studies, eg either in the first 2 weeks (noting could be 'well vaccinee effect' there, or possibly in a control period later after vaccination?

Response: Thank you for raising this – we agree that this population would be expected to have a high baseline rate of zoster, and that even the overall vaccine effectiveness of 50% would not be expected to be fully attained until towards the end of the 8 week period after vaccination for which we followed up. We consider the case ascertainment methods using both primary and secondary care records to be a strength of this study. These methods have previously been applied among the older general population and found to produce rates that were comparable to other published estimates for this age group, suggesting good case ascertainment of zoster using these methods. In response to both reviewers, we have expanded our discussion of the ascertainment of zoster cases in the discussion:

“It is possible that zoster disease may have been under-ascertained, either because patients did not attend healthcare or due to non-specific coding of zoster disease. A US study reported that 95% of patients aged over 60 years had attended healthcare when they experienced zoster disease,(7) and this might be expected to be higher among immunosuppressed individuals in a setting with universal healthcare. This study used both primary care and secondary care records to ascertain zoster cases, an approach which has previously been found to generate plausible estimates of zoster incidence among the older general population.(8) The sensitivity analysis was also designed to ascertain possible cases of zoster which may have been coded non-specifically as rash or encephalitis.”

We agree, it is possible that the small number of cases of zoster may reflect a healthy vaccinee bias – through clinical judgement as to whether and when to vaccinate. We have expanded our discussion to note that:

“Clinicians may have selectively, rather than inadvertently, vaccinated individuals at lower risk of vaccine-related disease, resulting in a ‘healthy vaccinee’ effect, and caution would be needed in generalising these findings outside of current practice in the context of guidance on contraindications.”

We have also taken the opportunity to highlight this point in the strengths and limitations box:

“Immunosuppression was not distinguished according to severity, but clinicians may have vaccinated selectively and caution would be needed in applying these findings outside of current vaccination practice.”

- pg 11, line 13, suggest say non-specific 'potentially' vaccine related disease

Response: Thank you, we have made this change.

- also noting in the discussion, that the most frequent contraindication was chemotherapy (? overall or in vaccinated), I wondered if more data in results on rates of inadvertent use by broad category of immunosuppression would be useful?

Response: Thank you, and we note that this useful suggestion was made by both reviewers. Our study was not designed to distinguish time at risk according to category of immunosuppression (which we address more fully above, in response to Reviewer 1's first comment). However, we agree with both reviewers that it would be a study question of interest, and a next step towards informed the current guidance on contraindications might be a study designed to examine who is vaccinated while immunosuppressed (by type of immunosuppression) and timing of vaccination relative to onset for relevant types of immunosuppression, preferably in a larger database for reasonably sized cohorts of patients with different types of immunosuppression.

Discussion

pg 13 lines 13-18 and Appendix 1A - its hard to tell which are 'maiden cohort's and which catch up. If 'maiden' have never had vaccine offered to them before, why would they contribute to decline in inadvertent use in later 2 study years.

Response: Thank you for highlighting this as needing clarification. The age-eligibility criteria for the 'catch-up' cohort changed throughout the study, and we hoped that Appendix 1A would help with this. We have added some explanation of which patients were considered maiden cohorts to the methods:

“Birth cohorts were defined as 'maiden cohorts' in the first year for which they were age-eligible for vaccination.”

We have expanded the discussion for a fuller consideration of why the apparent decline in inadvertent vaccination rates might be driven by a cohort effect among maiden cohorts, and hope that this possible explanation is now set out more clearly:

“In our study, analysis restricted to maiden years of eligibility suggests that the apparent decline in vaccination rates after year 2 is partly a cohort effect, whereby people who were unvaccinated despite previous eligibility were less likely to be vaccinated subsequently. This could be due to an initial decision not to vaccinate continuing over subsequent years of eligibility, or a greater focus on vaccination for newly eligible patients. Increasingly detailed guidance over time may also have helped reduce inadvertent vaccination.”

pg 13 line 25 - while I agree, with the statement for clinicians to consider vaccinating 'mildly immunosuppressed' individuals, it is in practice often difficult for primary care GPs/nurses to be sure of what is 'mild'. Are UK PHE changing their guidelines based on this study, or feel they can better define for clinicians what is mild versus not mild (noting current UK guidelines comprehensive but challenge is at individual patient level assessment)

Response: We agree that individual patient level assessment of immunosuppression is difficult, and find the results of our study reassuring that current practice appears to be safe, and even that there may be scope to adjust thresholds for contraindication. The findings in our study could reflect selective vaccination of some immunosuppressed individuals using clinical judgement, and we would be cautious to generalise from this study to a situation in which immunosuppressed individuals were routinely vaccinated. The researchers have recommended a review of the guidance on immunosuppression as a contraindication to live zoster vaccine but further research and wider clinical input may be needed to inform this. These findings will also be useful in consider the definitions, eligibility and priority immunosuppressed groups to receive the new recombinant vaccine. PHE are setting up an expert group to define groups who would be indicated to receive Shingrix and these data will inform that group.

pg 13 line 29 - pls provide reference for declining coverage data

Response: We have added a reference for the evaluation of the 5th year of the zoster vaccination programme (the 2017/18 national surveillance report), which shows that coverage has decreased over the 5 year programme: shingles vaccine coverage in the routine cohort (aged 70 years on 1 September 2017) was 44.4% in 2017/18 representing a 17.4% decline since the start of programme. The 2017/18 may have been a slight increase over the previous year once changes in eligibility criteria are taken into account, but the overall trend has been downwards and vaccine coverage at 44% remains suboptimal.

Appendix A2. Is it possible to list which (presumably many) biologics were included - this is common clinical question and some biologics have only narrow or v. little immunosuppressive effects

Response: Thank you for this helpful suggestion. We did include many biologics, as the guidance contraindicates vaccine for “those who are receiving or have received in the past 12 months biological therapy (e.g. anti-TNF therapy such as alemtuzumab, ofatumumab and rituximab) unless otherwise directed by a specialist” and we did not attempt to replace specialist judgement. We have listed the biologics in the table, as suggested.

VERSION 2 – REVIEW

REVIEWER	Steven A. Pergam Fred Hutchinson Cancer Research Center Seattle, WA, USA I have participated in clinical trials with Merck & Co.
REVIEW RETURNED	23-Dec-2019
GENERAL COMMENTS	The authors have addressed all my comments.
REVIEWER	Prof K Macartney NCIRS/University of Sydney, Australia
REVIEW RETURNED	23-Dec-2019
GENERAL COMMENTS	Revised manuscript and author responses satisfactory. However, language used to describe potential adverse events following immunisation with live attenuated zoster vaccine (LA ZV) is still not exactly correct - authors have replaced term 'shingles' with 'zoster disease' -but these are relatively synonymous - ie zoster or

	herpes zoster is reactivation of VZV from latency following PAST infection, not VZV-related disease associated with administration of LA-ZV - ie vaccine can't give you zoster, unless later post primary reactivation of Oka (vaccine strain) VZV. Please use correct term for vaccine virus associated disease which is VZV-related disease. Pedantic but important. 'Zoster' (ie dermatomal distribution) post vaccination, (as opposed to injection site or disseminated rash) is not a vaccine attributable adverse event, but vaccine failure.
--	--

VERSION 2 – AUTHOR RESPONSE

We are pleased to resubmit our manuscript with minor revisions, and we thank Prof Macartney for picking up on our consistency on this important point of terminology. We have amended the manuscript to refer to VZV-related disease when we are referring to VZV-related disease as a consequence of live vaccination. At certain points in the background and discussion we also use the term "vaccine-related disease" for clarity for general readers that we are referring specifically to disease that is caused by the live attenuated vaccine.

The manuscript now only uses the term "zoster disease" in the discussion when referring to other studies of disease caused by reactivation of VZV (such as a previous study in the US of healthcare attendance during episodes of zoster disease).